# Leveraging Physiologically Based Modelling to Provide Insights on the Absorption of Paliperidone Extended-Release Formulation under Fed and Fasting Conditions

**DOI:** 10.3390/pharmaceutics15020629

**Published:** 2023-02-13

**Authors:** Saima Subhani, Viera Lukacova, Chaejin Kim, Leyanis Rodriguez-Vera, Paula Muniz, Monica Rodriguez, Rodrigo Cristofoletti, Sandra Van Os, Elena Suarez, Stephan Schmidt, Valvanera Vozmediano

**Affiliations:** 1Center for Pharmacometrics and System Pharmacology at Lake Nona (Orlando), Department of Pharmaceutics, College of Pharmacy, University of Florida, Orlando, FL 32827, USA; 2Simulations Plus, Lancaster, CA 93534, USA; 3Model Informed Development, CTI Laboratories Spain, Derio, 48160 Bizkaia, Spain; 4Synthon BV, 6545 CM Nijmegen, The Netherlands; 5Pharmacokinetic, Nanotechnology and Gene Therapy Group (PharmaNanoGene), Department of Pharmacology, School of Medicine and Nursing, University of the Basque Country UPV/EHU, 48940 Bizkaia, Spain; 6Biocruces Health Research Institute, 48903 Bizkaia, Spain

**Keywords:** mechanistic absorption, food effect, regional absorption, physiologically based absorption modelling

## Abstract

Paliperidone was approved by the US FDA in 2006 as an extended-release (ER) tablet (Invega^®^) for the once-daily treatment of schizophrenia. This osmotic-controlled release oral delivery system (OROS) offers advantages, such as the prevention of plasma concentration fluctuation and reduced dosing frequency. The administration of the ER after a high-fat/high-calorie meal leads to increased maximum plasma concentration and area under the curve values by 60% and 54%, respectively. Food has various effects on gastrointestinal (GI) physiology, including changed transit times, changed volumes, altered pH in different GI compartments, secretion of bile salts, and increased hepatic blood flow. This may affect solubility, the dissolution rate, absorption, and the pharmacokinetics. The aim of this study was to apply physiologically based absorption modeling (PBAM) to provide insights on paliperidone ER absorption under fed and fasting conditions. The PBAM adequately predicted absorption from the OROS formulation under both conditions. Absorption primarily occurs in the ascending colon and caecum. After a high-fat/high-calorie meal, absorption is increased through the jejunum, ileum, and colon due to either increased solubilization or the better efficiency of the OROS technology. PBAM-guided approaches can improve the understanding of branded drugs and thereby aid in guiding the development of generic formulations or formulation alternatives.

## 1. Introduction

Food has numerous effects on gastrointestinal (GI) physiology, such as slow gastric emptying, increased hepatic blood flow, and prolonged GI transit times, and can thus affect oral drug absorption [1]. The amount and composition of the meal influence the rate of gastric emptying. Overall, the impact of food on GI physiology is a complex process that can have several effects on drug absorption, including (1) a delay in the absorption by reducing the gastric emptying rate, (2) a decrease in absorption, (3) an increase in absorption, or (4) no effect on the absorption [2]. The US Food and Drug administration (FDA) issued a guidance for the industry to streamline the design of food effect studies as well as fed bioequivalence studies to be conducted as part of drug development [3]. Comparative bioavailability (BA) studies in fed and fasted conditions (food effect studies) are recommended as part of new drug applications (NDA) and European marketing authorization applications. The recommendations include a two-period, two-sequence crossover study under fasting and fed conditions, preferably with healthy subjects. The meal used in the study should be a high fat (50%), high calorie meal (800–1000 calories) [3]. Furthermore, fed bioequivalence studies may also be recommended as part of abbreviated new drug applications (ANDA) and 10 (1) applications in Europe, i.e., for generic drugs [3]. In addition, there are certain potential implications of food on the development of modified-release drug products, such as dose dumping (which may result in adverse effects) and poor predictability of formulation-dependent effects of food from in vitro testing of the dosage form. Generic formulations can contain different inactive excipients and exhibit different release mechanisms than their corresponding reference drug product. In such cases, testing the possible effects of food on the generic formulation becomes essential as the interaction of food and the mechanism responsible for drug release from the dosage form in fed conditions is difficult to predict.

Paliperidone (PAL;9-hydroxy risperidone, the major metabolite of risperidone) is a second-generation antipsychotic that was approved as an extended release (ER) formulation (Invega^®^) for the once-daily treatment of schizophrenia by the US FDA in 2006. It is a central dopamine type 2 (D2) and serotonin type 2 (5HT_2A_) receptor antagonist [4]. PAL ER (Invega^®^) is generally well tolerated and adverse events are well studied. PAL offers several benefits over risperidone. These include but are not limited to renal excretion as the primary route of elimination and thus low risk for metabolic drug–drug interaction (DDI), as well as a decreased risk for extrapyramidal side effects due to differences in the affinity of the D2 and 5HT receptors. However, Invega^®^ is more expensive than (generic) risperidone, which resulted in cost-benefit concerns and the desire for less expensive generic PAL drug products. However, the development of generic PAL drug products is hindered by concerns around patent infringement of the OROS technology of the original product as well as the availability of cost-effective alternative formulations, which may be subject to altered food effects. Understanding the factors involved in the food effect of PAL’s OROS formulation will consequently serve as a valuable reference point when attempting to develop alternative, generic PAL drug products.

After administrating a single dose of Invega^®^, the maximum plasma concentration (Cmax) is reached approximately 24 h after dosing. The literature reported an absolute bioavailability value of PAL following an Invega^®^ administration of 28% [5]. The drug is categorized as a Biopharmaceutic Classification System (BCS) Class II compound (low solubility and high permeability). Vermier et. al., 2008, reported that one week after a single dose administration of 1 mg of [C^14^] PAL as an oral solution in five healthy males, 91.1% of the administered radioactivity was excreted: 79.6% in urine and 11.4% in the feces. The major route of elimination was reported to be renal elimination, with 59% of the dose of 1 mg oral solution excreted unchanged in urine. In addition, the study reported that PAL is the major circulating compound over a 24-h period when administered as a 1 mg oral solution [6]. Several biotransformation pathways, such as oxidative N-dealkylation, mono-hydroxylation, alcohol dehydrogenation, benzisoxazole, glucuronidation, and alicyclic hydroxylation, play very minor roles in the metabolism of PAL. In vitro studies suggest the roles of CYP2D6 and CYP3A4 in the metabolism of PAL, however, in vivo results suggest that these isoenzymes play a very limited role [6]. Moreover, in certain trans-epithelial Caco-2 transport studies using PAL at a concentration of 1 µM at physiologic pH, the estimated efflux ratio was 1.8, indicating active efflux transporters at this concentration [7]. Additionally, when the concentration increases, the efflux ratio gradually decreases. Finally, when the effect of the pH was studied, the facilitated absorption of PAL increases as the pH decreases by increasing the extraction ratio observed in the study. The above studies suggest the involvement of the efflux transporter in the active transport of PAL in the GI tract. However, not much information is available about the specific transporters involved and their kinetics.

After the administration of a single dose of PAL ER (12 mg dose) as Invega^®^ with food, there are 60% and 54% increases of the Cmax and area under the plasma concentration curve (AUC) compared to the fasting state [5,8]. Our primary focus was to use a physiologically based absorption modeling (PBAM) approach to improve the understanding on PAL ER absorption and the mechanisms involved in the formulation-dependent food effect using data from a pilot clinical study. In addition, the model was used to propose a dissolution profile more representative of the in vivo performance of the formulation that could be use as target to design biorelevant in vitro dissolution studies. This PBAM approach offers a great advantage and insights to help in the faster and more efficient development of ER generic formulations of the reference listed drug (RLD), Invega^®^.

## 2. Materials and Methods

### 2.1. Description of Dosage Form and Dissolution Data Input

Invega^®^ utilizes an osmotic, pressure-activated tri-layer tablet which delivers PAL at a controlled rate using osmotically controlled release oral delivery technology (OROS) [9]. The tri-layer system consists of two layers containing the drug and excipient and a push layer. In addition, there are two laser-drilled orifices on the drug layer side of the dome of the tablet. When the tablet is taken orally, it is expected that water enters through the semipermeable membrane that controls the rate at which water enters the tablet core (and thus the delivery rate) and hydrates the polymer forming the PAL-containing gel layer which is pushed out through the tablet delivery orifices [5]. A graphical representation of the release mechanism is presented in Figure 1.

Invega OROS^®^ marketed by Janssen Pharmaceuticals, Inc was bought from the US market. Considering water intake and release occur throughout the gastrointestinal tract, a pragmatic dissolution set up that would allow us to represent changes in the buffer pH as well as movement to agitate the gel layer was designed. A simple set up was defined to support quick development and potential applicability to other dosage forms and limit the number of factors that impact variability in the results. As the main driver of release is based on the osmotic push layer, it was not considered necessary to introduce different dipping speeds nor pH transitions for the dissolution test. In order to mimic the fasted state, the in vitro dissolution release profile from 6 mg Invega^®^ was evaluated in an assay under simulated gastric fluid (pH 1.2) and under fasting conditions for 2 h and then 22 h in USP phosphate buffer pH 6.8 at 15 dips per minute (DPM). Whereas in vitro dissolution release profile from 6 mg Invega^®^ under simulated fed conditions, was evaluated in acetate buffer (pH 4.5) for 2 h and then 22 h in USP phosphate buffer pH 6.8 at 15 dips per minute (DPM). The dissolution conditions consisted of 900 mL of media at 37 ± 0.5 °C using the Bio-DIS apparatus (reciprocating cylinder apparatus) . Aliquots were withdrawn at 0, 1, 2, 4, 6, 9, 12, 16, 20, and 24 h. All measurements were performed in triplicate.

### 2.2. Clinical PK Study Data

The clinical plasma concentration vs. time data after a single dose of 1 mg of PAL oral solution were obtained from the literature [6]. The PK data were available from two randomized, three-treatment, three-period, three-sequence, single-dose, comparative bioavailability pilot studies sponsored by Synthon, BV, comparing two different formulations of PAL 6 mg tablets (test) to 6 mg Invega^®^ tablets (reference) in healthy, adult male volunteers. One of these studies was conducted in fasting conditions, the other in fed conditions. There was a 7-day washout period between treatments in both studies. All volunteers were aged between 18 to 45 years with a body mass index in the range of 18.5 kg/m^2^ to 24.9 kg/m^2^, were provided written informed consent and were willing to follow the protocol requirements. Both study protocols were approved by the Drugs Controller General of India (DCGI) and by the corresponding Ethical Committees before the study started and were conducted according to the applicable Declaration of Helsinki. In each period, after a pre-dose fasting period of at least 10 h, the volunteers were administered a single oral dose of PAL (one of the two test formulations or the reference product, according to the randomization schedule) with 240 mL of water. The main difference between the studies was that in the fed state study, the dosing was preceded by a high-fat/high-calorie breakfast which was administered at 30 min before dosing, whereas no breakfast was provided to the volunteers in the fasting state study. Nine and ten evaluable volunteers were included in each study for the PK analysis. Plasma samples were analyzed using a validated HPLC-MS/MS method.

### 2.3. PBAM Model Development, Verification and Validation

Physicochemical and biopharmaceutical parameters were extracted from the literature or predicted based on the chemical structure with the ADMET Predictor^®^ module within GastroPlus^®^ (v9.8, Simulation Plus Inc., Lancaster, CA, USA). All parameter values (before and after optimization) are provided in Appendix A. The complete schematic flowchart used to develop and verify the models is depicted in Figure 2.

The different models built for PAL ER were internally verified and then further refined (if needed) with observations from the pilot clinical studies. The percentage prediction error (%PE) calculated using the following formulae was used to evaluate model prediction performance:Prediction error (PE) = (observed parameter − predicted parameter)/observed parameter × 100.

First, a compartmental PK model was developed using the PKPlus™ module (GastroPlus^®^ v9.8, Simulation Plus Inc., Lancaster, CA, USA) with data from the literature of an oral solution [6]. The objective was to set the disposition parameters of PAL before implementing the advanced compartmental absorption and transit (ACAT™) model (GastroPlus^®^ v9.8, Simulation Plus Inc., Lancaster, CA, USA). We tested different structural models (1, 2, and 3 compartments). The best model was identified based on the diagnostic plots, the Akaike information criterion (AIC), and a non-compartmental analysis of the predictions with the different models and observations. The ACAT model was then built on top of the compartmental model for the 1 mg solution using the fasted state physiology of humans. The model was externally verified by comparing simulated PK exposures with observations from a published study reporting PK data in 27 subjects receiving 2 mg of the oral solution [10].

Second, different processes involved in the drug release from the ER tablet (Figure 1) were accounted for by developing a model with the CR (control release) integral tablet formulation type. The default fasted physiology was used for setting the different ACAT™ parameters. The equations of the inbuilt Opt logD Model SA/V 6.1 model to adjust the absorption scaling factor (ASF) for each of the compartments of the small intestine and colon were used. The ASFs account for changes in passive transcellular permeability due primarily to changes in ionization and the distribution coefficient. The initial ACAT model used the concentration gradient version with the stomach transit time set to 0.25 h. Mean dissolution profiles (% release vs. time) were used as input in the model. Weibull parameters were optimized manually to the in vivo-observed plasma concentration profile and the triple Weibull parameter could give the best possible predictions. The Weibull maximum release was further manually fitted to get the best simulation. Also, the ascending colon transit time was increased by 4 h, to make the total gastrointestinal transit time up to 24 h, as observed from an OROS formulation. Then, simulations were set up for model verification using a single 6 mg dose and plasma concentration vs. time profiles were predicted for over a period of 96 h based on the design of the clinical pilot study under the fasted condition. Based on the comparison of the model predictions to the observations, the model was further refined by optimizing appropriate parameters using the optimization module in the software (whenever it was needed).

Third, the CR integral tablet formulation type was also used here to account for processes involved in the drug release from the ER tablet in the fed condition. The default fed physiology of the human was used for setting the different ACAT parameters. The default fed physiology accounted for increased bile salt concentration, increased stomach transit time, and pH (4 h, pH 4.9) compared to the fasted physiology. Weibull parameters were optimized manually to the in vivo-observed plasma concentration profile and the triple Weibull parameter could give the best possible predictions and was used as an input. The Weibull maximum release was further manually fitted to in vivo Cp vs. the time profile to get the best simulation. Furthermore, the ascending colon transit time was increased by 4 h, to make the total gastrointestinal transit time up to 24 h, as observed from an OROS formulation. Simulations of the PK profile during 96 h, mimicking the clinical study design by administering a 6 mg dose of PAL ER after the default meal, were performed for model verification.

Four, a local parameter sensitivity analysis was performed for the PBAM model for both the fasted and fed models with selected parameters (stomach transit time, caecum transit time, colonic transit time, C1, C2, C3, C4 (coefficient for the calculation of the absorption scaling factors for physiology model), the Weibull maximum release and reference solubility) using the in-built GastroPlus^®^ tool (v9.8, Simulation Plus Inc., Lancaster, CA, USA). The selection of the parameters was conducted based on the physiological implications, uncertainty around the ASF coefficients, and the drug/formulation characteristics.

Fifth, an external validation was performed using clinically relevant data from the literature. Available data after the single dose administration of paliperidone in fasted and fed conditions were included in the validation. Data included different doses in the range 3–15 mg and different ethnic populations (Caucasian, Chinese, and Japanese). The acceptance criterion was a predicted AUC_0–inf_ and Cmax within 2-fold of the observed.

### 2.4. Model Application: Virtual Population Simulations

Population simulations were performed to account for the high interindividual variability observed in the clinical pilot studies. For both fasted and fed conditions, 10 trials of N = 25 subjects each were simulated and compared with the observations. The average weight reported in the literature [11] for the Indian population was used in the simulations, as the specific demographics of the study subjects were not available. The population was set based on the default suggested %CV for most systemic and drug-related parameters except the %CV for total release (Max) was set to 20%. In addition, 10 trials of population simulations were conducted in 25 healthy subjects, randomly generated by GastroPlus^®^ (v9.8, Simulation Plus Inc., Lancaster, CA, USA). using the function of the virtual trial, in a crossover design, to compare the bio performance of the developed tablet. The %CVs for all parameters (drug and system-related parameters) were used as defaults in these simulations, except for the total release (Max), where the CV of 20% was used.

## 3. Results

### 3.1. Physiologically Based Absorption Model for the Oral Solution and ER OROS Formulation

A three-compartment distribution model best described the mean plasma concentration profiles of the 1 mg oral solution of PAL. The final PBAM model for the oral solution formulation constituted using the ACAT model on top of the three compartmental PK model is presented in the Appendix A. The model was able to adequately predict the observations with prediction errors in the PK metrics ≤ 18% (Appendix A). The external verification using an additional study from the literature after the administration of 2 mg of a PAL oral solution to N = 27 subjects [10] is provided in Appendix A. The PK metrics were well predicted with %PE < 19%.

The model for the ER in fasted conditions predicted mean plasma concentrations time courses after the 6 mg PAL ER in good agreement with the observations in the clinical study (Figure 3A).

The PBAM model for the ER formulation was then used to predict the PK under fed conditions (Figure 3B). The initial single simulation using the average subject resulted in a small shoulder peak within the first 15 h in the fasted and fed model predictions (Figure 3D) which was also present for the fasted state but was less remarkable (Figure 3C). When considering the amount dissolved, absorbed, and absorbed to the portal vein for both fasted and fed models in Figure 3C (fasted condition) and Figure 3D (fed condition), there was a disagreement between the amount dissolved and absorbed, thereby allowing for a gradual slowing of the amount absorbed across the GI tract, resulting in the appearance of the initial shoulder peak in concentration vs. the time profile of PAL. Therefore, C3 and C4 were increased (Appendix A, Figure 4A,B) and fitted to correct the slowing of absorption which is not a characteristic of the formulation and the drug. The final PBAM model for PAL ER under fasted and fed conditions was able to predict the observed plasma concentration profile reasonably well (Figure 4A and Figure 4B, respectively).

### 3.2. Prediction of PAL Regional Absorption from the Different Formulations

Figure 5 summarizes the application of the PBAM final models to investigate the regional absorption of PAL in the different compartments representing the GI tract and in the following three scenarios: (1) from the ER OROS formulation in a fasted state, and (2) from the ER OROS formulation in the fed state. In the case of oral solution, the model predicts absorption mainly occurring from the upper GI tract from the duodenum, jejunum, and ileum. However, in the case of PAL ER OROS, both under fasted and fed conditions, absorption occurs mainly in the ascending colon and caecum. However, when PAL is administered as with a high-fat/high-calorie meal, the percentage of drug absorbed increases resulting in an increase of the total amount from 35% to 52%. These values are in line with the reported oral bioavailability of Invega^®^ [5].

Moreover, the fed/fasted ratio of the mean of all the PK parameters (Cmax, AUC_0–t_, AUC_0–∞_) were greater than 1.25 and thereby results in a positive effect of food on PAL absorption.

### 3.3. Parameter Sensitivity Analysis (PSA) of the Developed Model

Local PSA was performed with selected model input parameters for both models under fed and fasted conditions. From the PSA results, the colonic transit time (C TransT) and total maximum Weibull release of the PAL ER OROS formulation resulted in being the most sensitive parameters for the fraction absorbed (Fa %) and Cmax. The PSA for these two parameters is depicted in Figure 6A,B.

Therefore, for further evaluation, the total max Weibull release was varied from 57% to 97% and ascending colonic transit time was kept at 14 h and several single simulations were carried out. It was observed that the exposure and Cmax increased with each increment in the total Weibull maximum release. However, the exposure and Cmax did not change beyond the ascending colonic transit time of 13.2 h. This information was used to further set the %CV for the population simulations used to capture variability in the observed data.

### 3.4. External Validation

The predicted-to-observed AUC and Cmax ratios were within twofold for all studies, meeting the validation criteria, and therefore the model was considered successful for further applications. Appendix A summarizes the results of the external validation.

### 3.5. Simulation Based Evaluation of PAL Variability in Fed and Fasted Conditions

The observed (geometric mean) and simulated plasma concentrations (geometric mean and 95% PI) from the 10 different simulated trials are shown in Figure 7A,B for the fasted and fed states, respectively. The overall geometric mean of the simulations (i.e., the geometric mean of the 10 different trials, red discontinues line) adequately predicts the mean of the observations (black dotted line).

### 3.6. Comparison of In Vitro Tested Dissolution and Weibull Deconvoluted Profiles Used as Input for Fed and Fasted Model

The developed final model for both the fasted and fed conditions used triple Weibull-derived parameters. The Weibull deconvoluted profiles representative of the in vivo dissolution profiles were plotted together with the experimental in vitro profiles using the specific dissolution media (fast- or fed-mimicking) under different stress conditions (5 or 15 dips per minute and 100 or 200 rotation per minute) (Figure 8).

The predicted in vivo differed from the in vitro profiles both in the shape of the curve and the maximum release, demonstrating the need for biorelevant media to generate in vitro profiles predictive of the in vivo behavior.

## 4. Discussion

We have developed and verified a minimal PBAM for the prediction of clinically observed food effects on PAL OROS formulation. This model brings forth the utility of PBAM to improve the mechanistic understanding of drug absorption and the impact of food effects by considering the physiological processes that can be affected by the food intake (such as increased liver flow, the effect of pH on the drug solubility, and the effect of bile salts in the gut and transit times) [12]. In the present study, the Caco-2 cell assay-predicted permeability value was used for PAL, and other essential inputs, such as pKa and log P, were used from the literature (Appendix A). Although PAL is a high-permeability drug, uncertainty exists around the role of efflux transporters on drug permeability and absorption. According to published in vitro permeability studies, there could be a role of efflux transporters in higher pH conditions; however, this is not well established, hence not included in the model [7]. The main limitation of the model is the unavailability of the OROS technology under the dosage form list within the software, and the CR integral tablet was chosen instead, where the unreleased material moves as a unit, i.e., all the unreleased drug remains in one compartment at any time, and the release is free to disperse. The OROS formulation system consists of an osmotically active tri-layer core containing two drug layers and a push compartment. After administration, water passes through and the hydrophilic polymers of the core hydrate and swell, creating a gel-like suspension containing PAL which is formed in situ and is then pushed out through the tablet orifices [12].

The present PBAM demonstrates that PAL ER from the OROS formulation is mainly absorbed in the distal part of the GI tract, in both fasted and fed conditions, and that the positive effect of food on drug absorption may be due to the increasing solubilization of the drug and/or due to higher colonic absorption in the case of the fed condition compared to the fasted condition [2]. It is important to consider that PAL is a low-solubility drug and that the colonic luminal fluid volume is low, i.e., it may be a rate-limiting factor for colonic absorption. Therefore, any factor that improves the solubilization of the drug, such as food intake, along with a higher colonic transit duration can increase oral bioavailability [13]. Another possible explanation is that the food effect can also be related to a difference in the release from the OROS formulation (and not necessarily solubilization). The mechanism of release requires a layer to take in water, swell, and push PAL-containing gel out into the GI tract. In fasting conditions, theoretically, the dosage form is emptied from the stomach faster than in fed conditions, where it should stay during the digestive cycle because it is not small enough to pass the sphincter with the liquid fractions that are emptied [1,2]. Staying more time in the stomach would provide the dosage form more time to take in water, which could result in a better functionality of the swelling layer resulting in more PAL being released and absorbed (higher “Weibull maximum release”) from the OROS dosage form in fed conditions than in fasting conditions. This is in line with the sensitivity analysis that identified the colonic transit time (C TransT) and total maximum Weibull release as the most sensitive parameters for Fa and Cmax. The longer colonic transit time would be translated into a longer time of the dosage form being capable of releasing in the colon. The model was able to adequately predict paliperidone OROS PK concentrations when the variability in the Weibull release was around 20%. However, one subject was identified as a physiological outlier in the fed study based on the population simulations and removed from the statistic calculations. The model was not able to capture the exposure of this subject, even when the most sensitive parameters, i.e., the maximum Weibull release and colonic transit time, were set to the maximum sensitive values (see Appendix A). On the other hand, the 20% variability in the maximum Weibull release was able to capture the subject with lower exposure under fed conditions, and was considered to have a formulation failure due to the behavior of the plasma profile.

Moreover, the model demonstrated that the in vitro dissolution profiles that are performed in non-bio-relevant conditions routinely are not predictive of in vivo dissolution and thus absorption. As the food effect may be related mainly to an increase in the solubilization of the drug, or a different Weibull maximum release, potential in vitro approaches to mimic this effect can be explored, such as the use of specialized buffers that represent fasting and fed conditions in non-sink conditions to assess the solubilization effect. Therefore, the use of biorelevant dissolution testing could be more predictive of the in vivo scenario [11,12,13,14].

During these last years, there has been an increased use of mechanistic models to study the effect of food on the PK of drugs [15,16,17,18,19]. With the present study, we have further demonstrated the applicability of model-based approaches to investigate the mechanisms involved in drug absorption and the effect of food on GI physiology, as well as the implications for in vivo food effects. Understanding the drug and formulation characteristics in conjunction with the physiology behind food effects is critical to assess if a food effect has the potential to be modulated through pharmaceutical approaches in order to adapt the drug product, or to determine if this is not possible according to the current galenical state-of-the-art, or to determine whether the impact must be assessed and managed clinically. The model can be further extended to the field of generic drug formulations of PAL to understand the absorption and food effect on PAL from ER formulations with different release mechanisms. A better understanding of the reference formulation and of the physiological and external factors such as the effect of food is fundamental to develop more efficient formulation development programs. For example, the design of alternative formulation concepts intended as non-patent infringing alternatives can be guided by further understanding of key physiological factors that impact release and absorption in vivo to ensure these aspects are preserved with the new formulation concept. On the contrary, when formulations are designed to improve aspects of release and absorption as part of a product development life cycle, understanding key factors allows us to define target, ideal formulation behavior in vivo. The PBAM model can be used to simulate the impact of what-if scenarios to limit the exploratory nature of initial formulation development in vivo and in vitro data generation. Applications of the model would include aiding the design of biorelevant in vitro dissolution studies and the design of pilot clinical trials. Once in vitro and in vivo data is generated with the new formulations, the PBAM model can be applied to increase knowledge of the absorption of the new formulations in fasted and fed conditions. Hence, the PBAM would increase the efficiency of the drug development program. In addition, once the model is validated with data from the pilot and pivotal studies, its application during the regulatory review processes could support the label information on dosing recommendations under fed and fasting conditions. If the final PBAM model is qualified with data from products with different quality attributes, it can be used to support product specifications and to justify the safe space for these attributes [20]. Establishing confidence in the model in iterative learning and confirming stages during the development of new formulations are key to impact the drug development process and demonstrate the utility of PBAM modeling as an efficient tool in industrial practice. A current challenge to routine application is that the approach must be rationalized per formulation concept, be both original and alternative, which requires teams with extensive understanding of biopharmaceutics, have in vitro in vivo relationships, a dissolution methodology, specific quality attributes, physiology, and use the equations and assumptions behind PBAM modelling, in the least. Currently, small pharmaceutical development teams may be challenged to incorporate all these aspects in their programs, but as the field evolves and may establish more standard approaches, a broader applicability to industry may be foreseen.

## 5. Conclusions

The mechanistic absorption modelling was integrated with in vitro dissolution testing to predict the regional absorption of PAL from Invega^®^ under fasting and fed condition with a high-fat/high-calorie meal. The ER models were developed and verified by incorporating abundant relevant in vitro and in vivo data and subsequently applied to predict the clinically observed effect of food on the exposure of PAL from Invega^®^. Additionally, PBAM and simulation improved our understanding of drug release and performance through a comprehensive view of the drug absorption processes in the GI tract and helped us to predict the in vivo release profile which can inform the selection of in vitro biorelevant media. It also enabled us to provide a rationale and a direction to apply these models so as to predict the effect of food on drug absorption by accounting for changes in the human physiology. These PBAM-guided approaches can greatly improve the understanding of branded drugs and thereby aid in guiding the development of generic formulations or formulation alternatives.

## Figures and Tables

**Figure 1 pharmaceutics-15-00629-f001:**
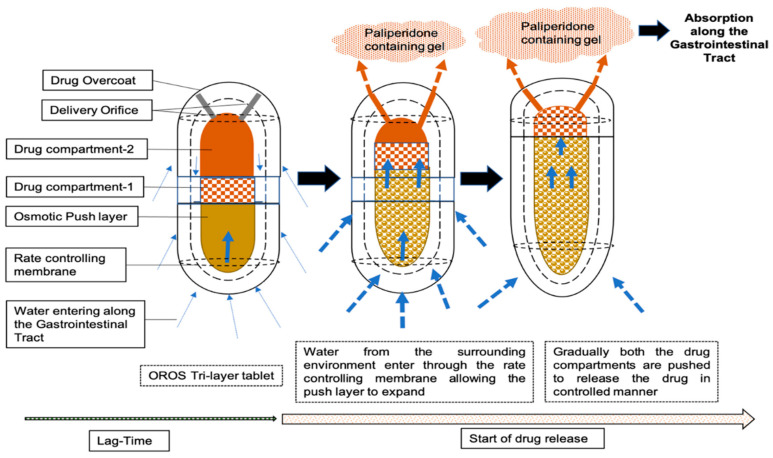
A schematic diagram describing the mechanism of release of PAL from tri-layer OROS formulation Invega^®^ for absorption along the GI tract.

**Figure 2 pharmaceutics-15-00629-f002:**
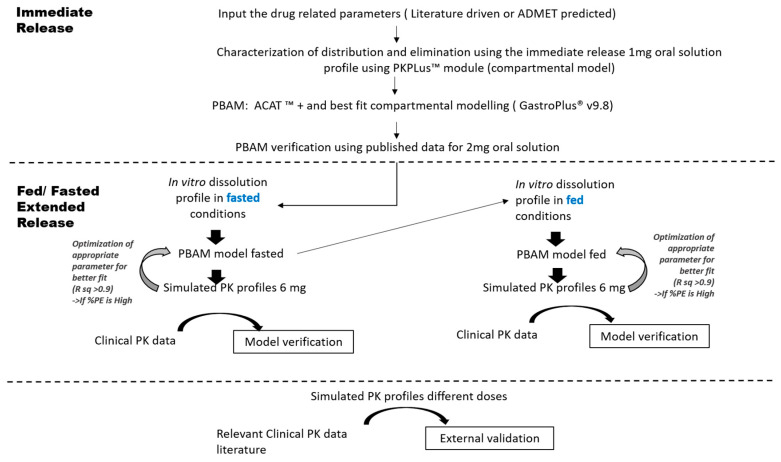
A schematic diagram describing the development and verification of the PBAM model for PAL under fasted and fed conditions. PBAM: physiologically based absorption modeling; PK: Pharmacokinetics.

**Figure 3 pharmaceutics-15-00629-f003:**
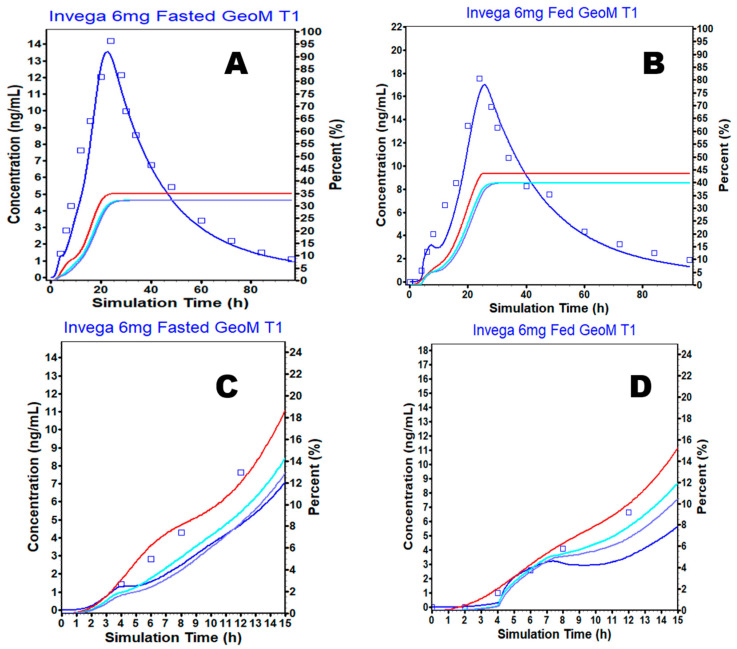
Predicted versus observed plasma concentrations over time after the oral administration of the PAL ER OROS tablet in (**A**) fasted and (**B**) fed (blue solid line—predicted plasma concentration, blue square box with bars—observed mean plasma concentration, red solid line—amount dissolved, purple solid line—amount to the portal vein, light blue solid line—amount absorbed). (**C**,**D**) are the same as (**A**,**B**) but with time truncated at 15 h to increase the resolution of the initial times.

**Figure 4 pharmaceutics-15-00629-f004:**
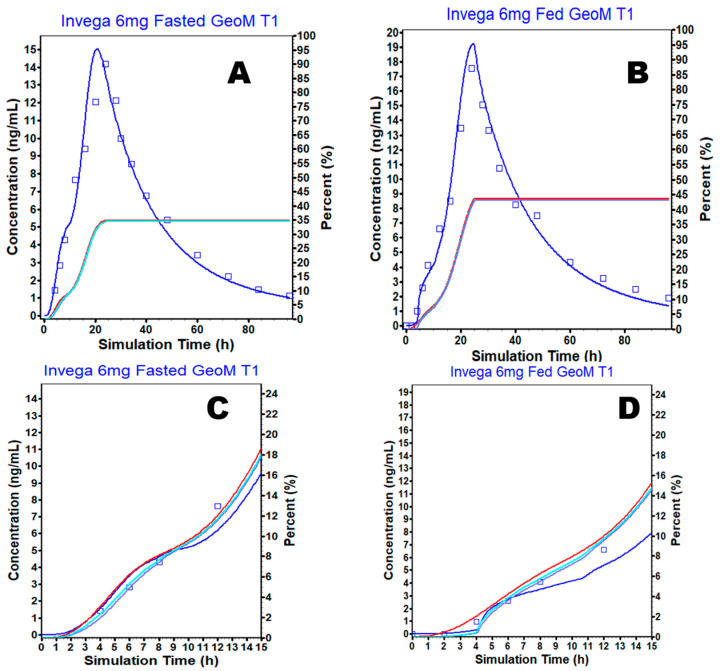
Predicted versus observed plasma concentrations over time using the final model after the oral administration of PAL ER OROS tablet in (**A**) fasted and (**B**) fed (blue solid line—predicted plasma concentration, blue square box with bars—observed mean plasma concentration, red solid line—amount dissolved, purple solid line—amount to the portal vein, light blue solid line—amount absorbed). (**C**,**D**) are the same than A and B but with time truncated at 15 h to increase the resolution of the initial times.

**Figure 5 pharmaceutics-15-00629-f005:**
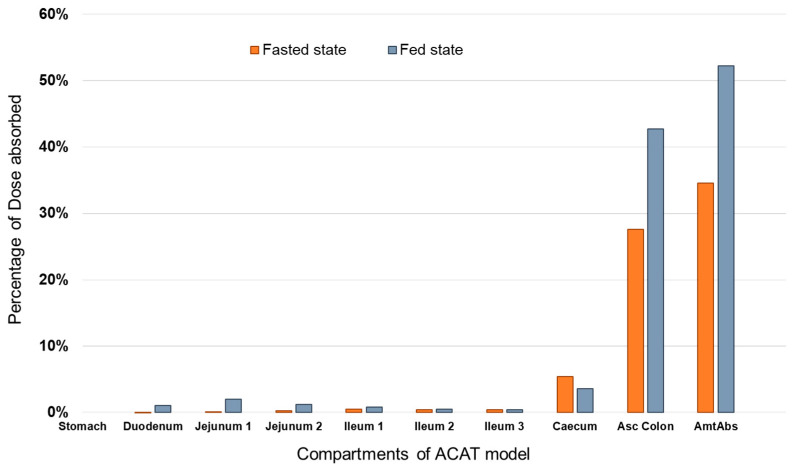
Bar chart representing the predicted regional absorption of PAL released from 6 mg PAL ER OROS tablet under fasted (orange bars) and fed (light-grey bars) conditions. AmtAbs represents the total amount absorbed of the drug for all GI compartments when taken together.

**Figure 6 pharmaceutics-15-00629-f006:**
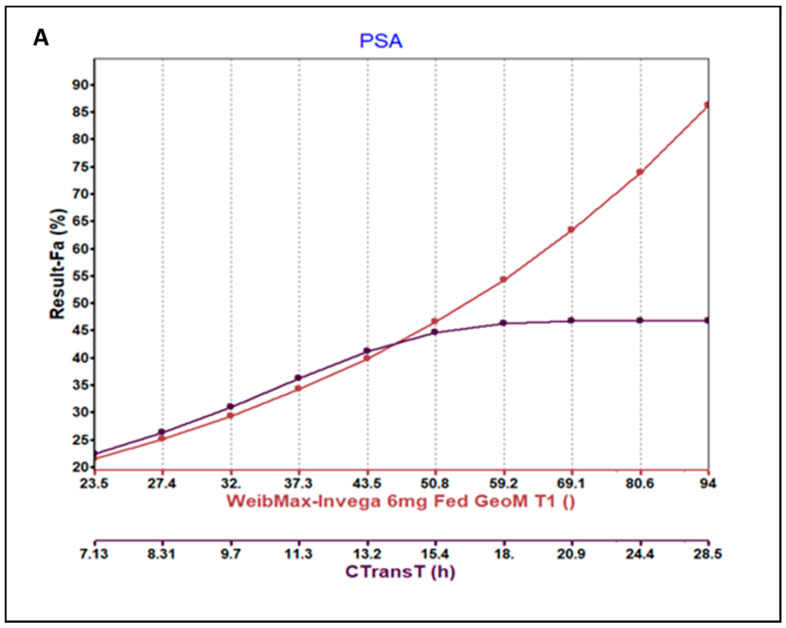
Parameter sensitivity analysis plot showing (**A**) fraction absorbed (%) vs. 2 axes: (a) red curve depicting changes in fed model-predicted Fa(%) for changes in total Weibull max release, and (b) black curve depicting changes in fed model predicted Fa(%) vs. ascending colon transit time (h). (**B**) Cmax (µg/mL) vs. 2 axes: (a) red curve depicting changes in fed model-predicted Cmax (µg/mL) for changes in total Weibull max release and (b) black curve depicting changes in fed model-predicted Cmax (µg/mL) vs. ascending colon transit time (h). PSA:parameter sensitivity analysis, CTansT: colonic transit time.

**Figure 7 pharmaceutics-15-00629-f007:**
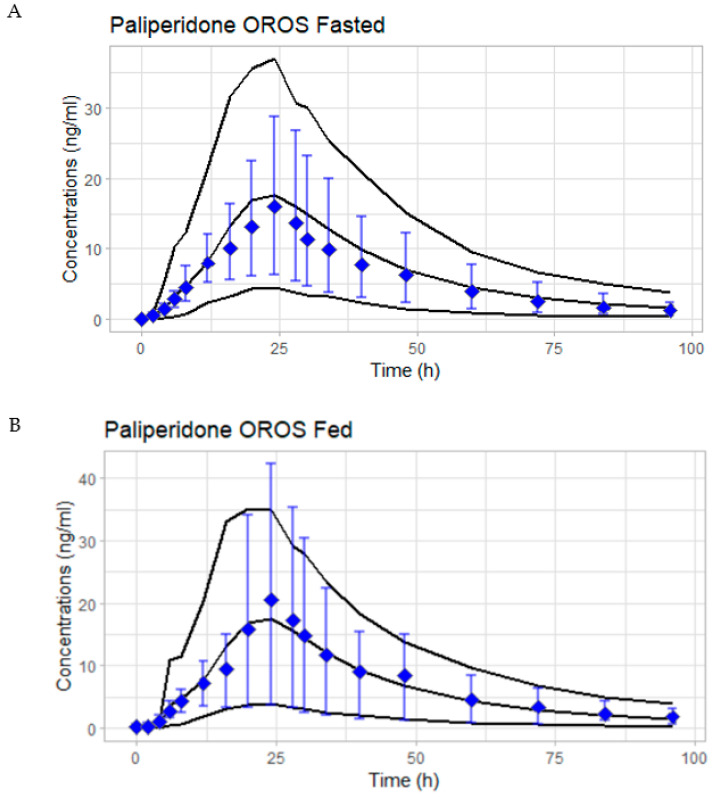
Population simulations of 10 different trials of N = 25 subjects each in fasted state (**A**) and fed state (**B**); plots are showing the simulated overall mean plasma concentrations of the 10 trials together and the 95% prediction bounds (black lines). The observed mean plasma concentration ± 95% bounds of the observations are presented using blue dots and bars.

**Figure 8 pharmaceutics-15-00629-f008:**
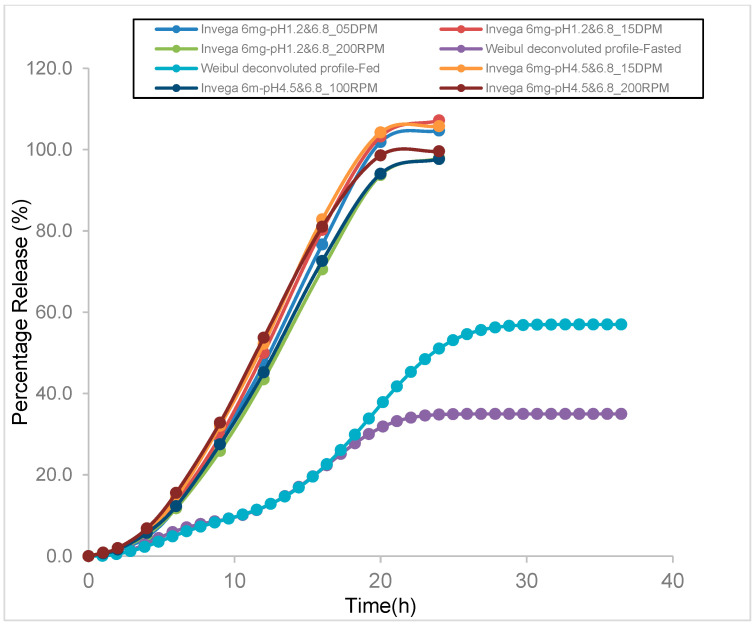
Percentage release (%) vs. time (h) of paliperidone ER OROS formulation (6 mg) in different fasting mimic dissolution media (pH 1.2 and pH 6.8) and fed mimic dissolution media (pH 4.5 and pH 6.8) under different conditions (5 or 15 dips per minute, 100 or 200 rotation per minute using BioDis apparatus).

## Data Availability

Not applicable.

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
