# Peer review of "Leveraging Physiologically Based Modelling to Provide Insights on the Absorption of Paliperidone Extended-Release Formulation under Fed and Fasting Conditions"

_pharmaceutics, 2023, doi:10.3390/pharmaceutics15020629_

Round 1
Reviewer 1 Report
Dear colleagues
The present manuscript titled with "Leveraging Physiologically Based Modelling to Provide Insights 2 on the Absorption of Paliperidone Extended Release Formula- 3 tion under Fed and Fasting Conditions" is well written and with novel data. Paliperidone is a golden drug for treatment of schizophrenia. The present study offers a great advantage and insights to help in the faster and more efficient development of Paliperidone.
- Please, add a conclusion of the study in the abstract sections.
Author Response
Please see the attachment.
Prof. Valvanera Vozmediano

Reviewer 2 Report
General: This is an important exercise in an evolving field of predicting drug absorption and food effect, especially for an ER formulation. However, the authors should present a realistic discussion of the results and conclusion, and need to update references in the discussion.
Introduction: A 2002 FDA Guidance is referenced, when there is a June, 2022 FDA Guidance available. Please update.
Methods: The word "validation" is never mentioned. If the model cannot be validated at the present time, this should be in the discussion and present what type of validation is necessary.
Discussion: The statement that PBAM "can" be used during drug development and regulatory review for food effect is an overstatement of the current situation and needs to be appropriately qualified. Also, references to mechanistic models of food effect contains old references when multiple specific recent examples would be more appropriate. Additional discussion about the steps necessary to get us from our present knowledge to the more routine use of PBAM in drug development would be welcomed.
Author Response
Please see the attachment
Prof Valvanera Vozmediano

Reviewer 3 Report
The manuscript's authors investigate an osmotic drug delivery system that has been commercially available for a long time. In the case of the preparation containing paliperidone, the aim was to develop a physiologically based absorption modeling (PBAM) approach to improve the understanding of the absorption of the active ingredient and the mechanisms involved in the food effect dependent on the preparation, also based on the data of a pilot clinical study. The aims of the manuscript are clear, the text is well edited, and the figures are informative, but some corrections and additions are necessary before publication.
1. The manufacturer of the drug product is not indicated in the manuscript
2. The authors wrote in line 125 that the patients orally take the tablet, and the water enters the drug delivery systems through the orifice of the drug product and hydrates the polymer. According to the information provided by the 5th referent and the OROS system, this is not the case. Water also enters through the semi-permeable membrane. The sentence must be changed.
3. There are several shortcomings in the description of the in vitro dissolution test. The type and manufacturer of the Bio-Dis equipment are not indicated in the text. Mesh sizes (bottom and top) are missing for the dipping tube. The applied temperature, dissolution medium volume, sampling times, and sample volume are also missing. The number of parallel measurements is not specified either.
4. It would be essential to specify the analytical method used to determine the active ingredient content during the in vivo clinical and in vitro dissolution tests.
Author Response

(The authors gave the same response as above.)

Reviewer 4 Report
The manuscript entitles “Leveraging Physiologically Based Modelling to Provide Insights on the Absorption of Paliperidone Extended Release Formulation under Fed and Fasting Conditions” applied a physiologically based absorption model (PBAM) to provide insights on paliperidone extended-release (ER) tablet absorption under fed and fasting conditions. The PBAM model adequately predicted that absorption primarily occurs in the ascending colon and caecum and a high-fat/high-calorie meal leads to an increase of absorption due to either an increased solubilization or better efficiency of the OROS technology. This manuscript provides a reference to application prospects of the paliperidone extended release formulation in attempting to develop alternative generic PAL drug products.
However, I still have some incomprehensible points which need to be answered by the authors.
1 What was the basis for the research method adopted in "2.1. Description of Dosage form and resolution data input"? reference?
2 There are pharmaceutical preparations of different specifications (3, 6, 9mg). Why did the author use 6mg for the study?
3 Why was pH 4.5 under dietary conditions in line 136?
4 The information about GastroPlus® version 9.8 soft was not enough. The key parameters of the modeling process are lacking.
5 What were the reasons for the unavailability of specific demographic data for the study subjects in line 231?
6 Was 1mg or 6 mg dose used in this study?
7 In Figure 3, what was the reason for the sudden increase of drug concentration in the diet state? No explanation in PBAM model. Why did this happen in the fourth hour?
8 As shown in Figure 8, the points with different colors were not marked.
9 What was the specific significance of this manuscript to guide generic PAL drug products?
Author Response

(The authors gave the same response as above.)

Round 2
Reviewer 3 Report
Dear Authors, Dear Editor!
Thanks for the answers, I agree with them, and I accept them. I have no further questions about the manuscript.